# Climate-Driven Adaptive Differentiation in *Melia azedarach*: Evidence from a Common Garden Experiment

**DOI:** 10.3390/genes13111924

**Published:** 2022-10-22

**Authors:** Boyong Liao, Qingmin Que, Xingming Xu, Wei Zhou, Kunxi Ouyang, Pei Li, Huaqiang Li, Can Lai, Xiaoyang Chen

**Affiliations:** 1College of Horticulture and Landscape Architecture, Zhongkai University of Agriculture and Engineering, Guangzhou 510225, China; 2College of Forestry and Landscape Architecture, South China Agricultural University, Guangzhou 510642, China; 3Guangdong Key Laboratory for Innovative Development and Utilization of Forest Plant Germplasm, Guangzhou 510642, China; 4China Forestry Group Leizhou Forestry Bereau Co., Ltd., Zhanjiang 524043, China

**Keywords:** common garden, *Q*
_ST_, *F*
_ST_, *Melia azedarach*, temperature, local adaptation

## Abstract

Studies of local adaptation in populations of chinaberry (*Melia azedarach* L.) are important for clarifying patterns in the population differentiation of this species across its natural range. *M. azedarach* is an economically important timber species, and its phenotype is highly variable across its range in China. Here, we collected *M. azedarach* seeds from 31 populations across its range and conducted a common garden experiment. We studied patterns of genetic differentiation among populations using molecular markers (simple sequence repeats) and data on phenotypic variation in six traits collected over five years. Our sampled populations could be subdivided into two groups based on genetic analyses, as well as patterns of isolation by distance and isolation by environment. Significant differentiation in growth traits was observed among provenances and families within provenances. Geographic distance was significantly correlated with the quantitative genetic differentiation (*Q*_ST_) in height (HEIT) and crown breadth. Climate factors were significantly correlated with the *Q*_ST_ for each trait. A total of 23 climatic factors were examined. There was a significant effect of temperature on all traits, and minimum relative humidity had a significant effect on the survival rate over four years. By comparing the neutral genetic differentiation (*F*_ST_) with the *Q*_ST_, the mode of selection acting on survival rate varied, whereas HEIT and the straightness of the main trunk were subject to the same mode of selection. The variation in survival rate was consistent with the variation in genetic differentiation among populations, which was indicative of local adaptation. Overall, our findings provide new insights into the responses of the phenological traits of *M. azedarach* to changes in the climate conditions of China.

## 1. Introduction

Members of the genus *Melia* Linnaeus (Meliaceae: Melioideae: Melieae) are deciduous trees or shrubs that occur in tropical and subtropical areas of the eastern hemisphere, including south of the Yellow River in China. The genus *Melia* has been proposed to comprise either one (*M. azedarach* Linn.) or three (*M. azedarach*, *M. toosendan* Sieb. et Zucc., and *M. dubia* Cav.) species [1,2,3,4], and much debate remains regarding the taxonomy of this group. *Melia* species can grow rapidly, and they are widely grown in China, especially *M. azedarach*, because of the mechanical properties of their wood, which makes them resistant to insects, pathogens, and corrosion [1]. *Melia* species are also often used as medicinal plants for the treatment of various diseases, including diarrhea, diabetes, rheumatic diseases, and hypertension [5]. Several studies have examined the chemical constituents (e.g., limonoids and triterpenoids) extracted from the fruits, stem bark, and leaves of *M. azedarach* and *M. toosendan* [6,7]. Plants in the genus *Melia* might provide useful future medicines in light of their diverse medicinal properties.

The features often used to distinguish *Melia* species, such as the genes underlying differences in phytochemicals and the phylogenetic relationships among species, have not been thoroughly explored. This deficiency in our knowledge of the features delimiting these species affects the efficiency of genetic breeding programs, as well as the ability of *Melia* species to be used as medicinal plants. Morphological differences among *Melia* species are subtle. *M. azedarach* has the following distinguishing features: 3–6-loculed ovaries, small fruits less than 2 cm in length, leaflets with blunt teeth, and inflorescences as long as the leaves. *M. toosendan* has the following distinguishing features: 6–8-loculed ovaries, large fruits approximately 3 cm in length, leaflets with indistinct blunt teeth on the entire margin, and inflorescences half as long as the leaves. *M. dubia* has the following distinguishing features: petals pilose on both sides, elliptic drupes, and distribution restricted to south of the Five Ridges. *M. toosendan* is often considered a synonym of *M. azedarach*, and it has been infrequently documented in China [1,4]. The geographic distributions of *M. azedarach* and *M. toosendan* overlap; they also show overlap in their flowering time (April to May for *M. azedarach* vs. March to April for *M. toosendan*). These species were likely generated by a sympatric speciation event, and they might naturally hybridize given that their flowers are diecious. Populations of the genus *Melia* in China thus represent a *M. azedarach* species complex; *M. azedarach* is considered the model species of the genus *Melia* [1].

Local adaptation occurs when the mean fitness of a population is greater in its native habitat than in foreign habitats [8,9,10]. Local adaptation is shaped by various evolutionary mechanisms, including selection, genetic drift, mutation, and gene flow. Divergent selection often acts on populations occurring in spatially heterogeneous environments; local adaptation to such environments can thus contribute to shaping the responses of populations to changes in the environment [11]. Several methods have been used to study local adaptation. Field experiments can be conducted to clarify the role of environmental factors (i.e., non-genetic factors) underlying phenotypic differentiation among populations and at various geographic scales [12,13]. Common garden experiments can also be conducted to clarify the relative importance of different factors in shaping local adaptation between populations using comparisons of neutral genetic differentiation (*F*_ST_) and quantitative genetic differentiation (*Q*_ST_); genome scans can be used to identify genes involved in local adaptation, and genomic regions under selection can be identified in populations via high-density genetic markers [14,15,16]. Linkage maps and QTLs have been used to reveal fitness-associated phenotypic variation that is closely related to genotypes linked with marker loci; this approach has also been used to generate linkage maps and find association sites in species with large populations, short life cycles, and abundant molecular markers, such as Arabidopsis and rice [17,18,19]. Genome-wide association studies can also be used to identify single-nucleotide polymorphisms and alleles associated with environmental factors and phenotypes in species with several populations, large sample sizes, and reference genomes [20,21].

The provenance trial approach is generally used to (i) characterize variation in quantitative traits among populations, (ii) determine progeny performance, and (iii) estimate genetic parameters [22]. This approach has been used to study several tree species, including *Platycladus orientalis* [23], *Populus fremontii* [24], *Populus trichocarpa* [25], *Eucalyptus dunnii* [26], and *Pinus caribaea* [27]. Provenance trials are also effective for detecting natural selection and predicting the ability of populations to adapt to future climate change [28,29,30,31]. Several studies have shown that provenance trials can be used to characterize local adaptation among populations and clarify differentiation in quantitative traits among populations [32,33]. Previously conducted provenance trials of the *M. azedarach* complex have focused on estimating heritability and predicting genetic gain to aid breeding programs [34,35,36]. However, the local adaptation of various populations in the *M. azedarach* complex, as well as the role that lineage sorting has played in mediating local adaptation, has not yet been clarified. Here, we used provenance trials over five years to determine the role that selection has played in driving differentiation among populations of the *M. azedarach* complex. The results of this analysis provide key information that will aid forest breeding programs.

Simple sequence repeat (SSR) markers are generally considered selectively neutral, and an analysis of SSRs can provide information complementary to data on adaptive molecular variation among populations [37]. Specifically, this analysis involves characterizing (i) the population genetic structure to evaluate the rates of natural hybridization; (ii) the effects of isolation by distance (IBD) to quantify the levels of gene flow; and (iii) the evolutionary genetic relationships among populations to evaluate their taxonomic status. A comparison of the population differentiation indicated by molecular markers ($F_{st}$) [38] and that driven by quantitative traits ($Q_{st}$) [39] can be used to identify the relative importance of climate factors and geographic distance in shaping the evolution of the traits mediating adaptation. Thus, data from provenance trials and information on population history can be used to clarify patterns of differentiation in the *M. azedarach* complex.

The aims of this study were to determine (i) whether there is local adaptation among *M. azedarach* populations according to analyses of phylogenetic relationships and population structure derived from molecular markers, and (ii) whether ecological adaptation plays a role in species differentiation through provenance trials. The results of these analyses will provide new insights into the role of natural selection in shaping patterns of differentiation among *M. azedarach* populations.

## 2. Materials and Methods

### 2.1. Sample Sites and Collection of Seeds

Seed samples were obtained from 43 populations across the entire range of the *M. azedarach* complex (Table 1; Figure 1). These samples were obtained as far south as a tropical rainforest in southern China (Lingshui, Hainan province) to as far north as a temperate monsoon forest in northern China (Tai’an, Shandong province); the difference in the mean annual temperatures between these regions is 16.4 °C. Samples of putative *M. toosendan* and *M. azedarach* could not be distinguished according to seed size and leaf morphology. *M. dubia* samples could not be identified because the sizes of their seeds and other distinguishing leaf traits remain unclear. The number of samples obtained from each population ranged from 14 to 15 for the SSR dataset; in each provenance trail, each population comprised 1–26 families with 50 individuals each (Experiment I, Table 1). Provenance trials were conducted (Experiment II, Table 1).

A nuclear SSR marker analysis was conducted using 31 samples, including ten populations with half-sib pedigrees and 21 populations lacking known pedigrees (Table 1). Samples used in the SSR marker analysis were from populations spanning the entire range of the *M. azedarach* complex (Figure 1 and Table 1).

### 2.2. Provenance Trials

Seedlings from the 22 provenances, including 236 half-sibs, were obtained from seeds sown in a field at the South China Agriculture University, Guangzhou, Guangdong province, China, in February 2014 (Table 1). When the seedlings were 30 cm in height, 50 individuals per family were transplanted to the Leizhou Experimental Site at Suixi, Zhanjiang (20°57′12″ N, 109°48′34″ E), Guangdong province, China. This site is a subtropical area featuring a maritime monsoon climate. The average annual temperature is 23.5 °C, and the average annual precipitation ranges from 1417 to 1804 mm. The rainy season runs from May to October. Approximately two typhoons make landfall at this site annually.

Our field experiment was conducted using a randomized block design with 10 replicate blocks and five plots (each corresponding to a single family) per block in 3 m intervals, with a spacing of 1.5 m between plots. A total of 11,800 trees were planted on 1 May 2014.

### 2.3. Molecular Procedures

Leaf samples were obtained from 31 populations (Table 1) and stored at 80 °C until DNA extraction. The E.Z.N.A. high-performance DNA Mini Kit (Omega Bio-Tek, Inc. Norcross, GA, USA) was used to extract DNA from 150 mg leaf samples; the DNA was separated using electrophoresis on a 1.0% agarose gel. The concentration of DNA was measured using a Nano-Drop 1000 spectrophotometer (Thermo Fisher Scientific, Waltham, MA, USA); after adjusting the concentration to 50 ng/µL, the DNA was stored at 20 °C until PCR amplification.

A total of 15 SSR markers were screened from tests of 135 primer pairs for the *M. azedarach* complex (Appendix A, [40]). PCR reactions were conducted in a total volume of 15 μL with approximately 50 ng of DNA, 1× PCR buffer, 1.33 mM MgCI_2_, 8 μM each pair of primers, 0.67 mM dNTPs, and 0.75 U Taq Polymerase (Invitrogen Inc., Carlsbad, CA, USA). The reactions were conducted in an Eastwin Thermal Cycler (EDC-810, Suzhou, China) with the following thermal cycling conditions: 4 min at 94 °C; 21 cycles of 94 °C, annealing at 56 to 60 °C for 30 s, with the annealing temperature decreased by 0.5 °C in each cycle, and an extension at 72 °C for 1 min; 25 cycles of 94 °C for 30 s, annealing at 52 to 62 °C for 30 s, with the annealing temperature decreased by 0.5 °C each cycle, and an extension at 72 °C for 1 min; and a final extension for 10 min. The samples were then stored at 4 °C for genotyping. Polymorphic bands of SSR markers were detected using silver staining methods via 6% denaturing polypropylene gel electrophoresis [41].

### 2.4. Analysis of Data from the Provenance Trials

We measured the following growth traits of *M. azedarach* for five consecutive years from 2015 to 2019: maximum stem height (HEIT), diameter at breast (1.3 m) height (DBH), the ground diameter in the first year (GBH), and the crown breadth (CRB). Measurements were also taken on the straightness of the main trunk (SMT), the number of branches (NOB), and the clear bole height (CBH) in 2016. Measurements of the survival rate (SR) were taken every year.

The straightness of the main trunk was scored visually using a scale from 1 to 4, with 1 indicating a trunk that was straight, 2 indicating a trunk that was slightly bent, 3 indicating a trunk that was largely bent, and 4 indicating a trunk with many twigs emanating from the ground. We used HEITi to refer to the maximum stem height (HEIT) at the *i*th measurement, with *i* = 1, 2, 3, 4, and 5 corresponding to measurements taken in 2015, 2016, 2017, 2018, and 2019, respectively. Other traits were referred to using a similar notation. A total of 30 traits were measured for five consecutive years (Appendix A).

Each observation was described using a mixed linear model:*Y_ijkl_* = *μ* + *B_i_ + P_j_* + *F*_*k*(*j*)_ + *E_ijkl_*(1)
where *Y_ijkl_* is the *l*th observation in the *k*th family of the *j*th provenance in the *i*th block, *μ* is the overall mean, *B_i_* is the fixed effect of the *i*th block (*i* = 1, 2,…, 10), *P_j_* is the random effect of the *j*th provenance (*j* = 1, 2… 22), *F_j_*_(*i*)_ is the random effect of the *k*th family nested in the *j*th provenance, and *E_ijkl_* is the residual error.

Population genetic differentiation associated with quantitative traits was quantified using *Q*_ST_ [39,42,43], which was calculated as follows:(2)$$Q_{\mathrm{ST}}=\frac{\sigma_{P}^{2}}{\sigma_{P}^{2}+8\sigma_{F}^{2}}$$
where $\sigma_{P}^{2}$ is the variance in provenance effects and $\sigma_{F}^{2}$ is the variance in family effects. The variance components of $Q_{\mathrm{ST}}$ were calculated for each population pair (231 pairs, Appendix A) and each trait using REML [44] in the R package AsReml-R 4.1.0.106 [45].

Climate data were used to characterize the environmental conditions at each provenance site and assess the effect of ecological variation on patterns of population differentiation. A total of 23 climatic variables were obtained from the National Meteorological Information Center of China (http://data.cma.cn/, accessed on 6 July 2016). These monthly climate data were collected from 22 weather stations in China from 1951 to 2012, and the means of each climatic factor were calculated (Appendix A). A pairwise distance matrix of populations was generated based on the absolute difference in climate variables between populations. Pearson correlation coefficients were estimated between the distance in each climate variable and the magnitude of the population differentiation in each trait.

### 2.5. Analysis of SSR Markers

A micro-checker was used to detect null alleles for all the genotypes obtained from 15 SSR primer pairs [46]. Analyses of molecular variance, *G*st (unbiased *F*st), Mantel tests, and allele frequency estimates were obtained using GenAlEx v. 6.502 [47]. Linkage disequilibria between loci and *F*_ST_ were estimated at multiple loci using Genepop v4.7 [48].

Genetic relationships among populations were inferred using Nei’s genetic distance [49] and the neighbor-joining method with 1000 bootstrap replicates in Phylip 3.695. Phylogenetic relationships were visualized using Poptree2 [50]; the Dsw distance [51] was calculated using microsatellite DNA data. An analysis of the population structure was conducted using the Structure 2.3.4 [52] software with the admixture model and assuming that the allele frequencies were correlated. The parameters in the population structure analysis were as follows: 10 iterations from K = 1 to K = 10 with 100,000 burn-in iterations and 100,000 iterations after the burn-in period, which ensured a steady-state distribution for parameter estimation. STRUCTURE HARVESTER (http://taylor0.biology.ucla.edu/structureHarvester/, accessed on 5 December 2015), [53]) was used to determine the optimal number of clusters (K-value).

To evaluate the relative importance of IBD, the geographic distance between populations was computed using the spherical distance (*L*) formula based on the longitude, latitude, and elevation:(3)$$L=\sqrt{{\left(x_{1}-x_{2}\right)}^{2}+{\left(y_{1}-y_{2}\right)}^{2}+{\left(z_{1}-z_{2}\right)}^{2}}$$
where $x_{i}=\left(h_{i}+R\right)\times cos(l_{a\left(i\right)})\times cos(l_{o\left(i\right)})$; $y_{i}=\left(h_{i}+R\right)\times cos(l_{a\left(i\right)})\times sin(l_{o\left(i\right)})$; $z_{i}=\left(h_{i}+R\;\right)\times sin(l_{a\left(i\right)})$; $l_{a\left(i\right)}$, $l_{o\left(i\right)}$, and $h_{i}$ are the longitude, latitude, and elevation of the ith population (*i* = 1, 2), respectively, and *R* is the radius of the Earth. Regression analysis [54] and Mantel tests (10,000 permutations) were performed between the matrices of geographic distance (Appendix A) and population differentiation using *R* scripts.

Differences between *F*_ST_ and *Q*_ST_ were tested using the R package QstFstComp (https://github.com/kjgilbert/QstFstComp, accessed on 11 April 2019), [43]) to assess the local adaptation in quantitative traits. The bootstrapping method [55] was used to compute confidence intervals for *Q*_ST_ and correct for variable sample sizes [43]. In this approach, observed differences between *Q*_ST_ and *F*_ST_ were compared against the simulated expected distribution under a neutral hypothesis. Significant differences indicated the presence of natural selection [42].

## 3. Results

### 3.1. Population Genetic Differentiation

Null alleles were absent at all 15 SSR loci in the 31 populations according to the micro-checker analysis. A total of 110 of the 465 alleles (23.65%) were not in a Hardy–Weinberg equilibrium (HWE) after a Bonferroni correction (α = 0.05/465 = 0.00011) (Appendix A). The inbreeding coefficients (*F*_IS_) were generally negative at individual loci, and the observed heterozygosity was not greater than the expected values under a HWE.

The average polymorphic information content (PIC) was approximately 73.9%; the highest PIC value was observed in Liuyang, Hunan province (77.3%) and the lowest PIC value was observed in Wuzhishan, Hainan province (66.6%) (Table 2). Shannon’s information index (*I*) was highest (1.81 ± 0.46) in Ceheng, Guizhou province, and *I* was lowest (1.46 ± 0.43) in Wuzhishan, Hainan province; the average value of *I* was 1.69 ± 0.39. The number of observed alleles ranged from 6.00 ± 2.93 (Wuzhishan, Hainan province) to 8.60 ± 3.44 (Ceheng, Guizhou province), and the average number of observed alleles was 7.27 ± 2.46. The number of effective alleles ranged from 4.09 ± 1.86 to 5.78 ± 2.62, and the average number of effective alleles was 5.05 ± 1.85. The mean expected heterozygosity (He) ranged from 0.74 ± 0.11 (Wuzhishan, Hainan province) to 0.83 ± 0.06 (Liuyang, Hunan province), and the average He was 0.80 ± 0.08. The He in each population was greater than the average value of Nei’s gene diversity.

### 3.2. Genetic Diversity according to G_ST_ and IBD

An analysis of the population genetic differentiation revealed that 8.6% of the total genetic variation at multiple SSR loci was distributed among populations. The genetic differentiation significantly differed from zero at individual loci (*p* < 0.001; Table 3), and the *G*_ST_ values ranged from 0.0327 at the SSR120 locus to 0.1281 at the SSR117 locus. Significant relationships between the inverse number of migrants between populations inferred from *G*_ST_/(1 − *G*_ST_) and the logarithm of the geographic distance were observed for the loci SSR54, SSR74, SSR111, SSR117, and SSR123 (Table 3). A significant but weak effect of IBD was observed according to a multilocus analysis (a = −0.0353 and b = 0.0063; *p* = 0.010; R^2^ = 0.06; Table 3).

The *F*_ST_/(1 − *F*_ST_) values ranged from 0.034 to 0.242, and the natural logarithm of the geographic distance ranged from 10.28 to 14.51. Mantel’s test based on multilocus *F*_ST_/(1 − *F*_ST_) and geographic distance matrices revealed weak IBD effects at multiple loci (r = 0.245, *p* = 0.01, Figure 2).

### 3.3. Genetic Structure and Genetic Relationships among Populations

The STRUCTURE analysis revealed a maximum Δ*K* value at K = 2 (Δ*K =* 50), which indicated that the 31 populations could be divided into two groups (Figure 3). Group I (Figure 4, K = 2 in red) comprised populations 739 and 754 from the Yunnan province; populations 842, 843, and 858 from the Guizhou province; populations 1060 and 1061 from the Sichuan province; and population 1565 from the Gansu province. Group II (Figure 4, K = 2 in yellow) comprised population 102 from the Fujian province; populations 205 and 248 from the Jiangxi province; populations 307, 308, and 310 from the Hunan province; populations 412 and 415 from the Guangdong province; populations 524 and 525 from the Hainan province; populations 628, 631, and 652 from the Guangxi province; population 741 from the Yunnan province; population 844 from the Guizhou province; population 959 from the Zhejiang province; population 1162 from the Anhui province; population 1363 from the Henan province; population 1464 from the Shannxi province; population 1666 from the Hebei province; populations 1767 and 1768 from the Shandong province; and population 1869 from the Hubei province.

When the populations of Group I and Group II were mapped to their geographic locations (Appendix A), Group II could be separated into two subgroups, with nine populations (248, 959, 1162, 1363, 1464, 1666, 1767, 1768, and 1869) from the middle and lower reaches of the Yangtze River to downstream of the Yellow River, and the other fourteen populations from the southern side of the Yangtze River to Hainan Island.

Nei’s genetic distance between populations ranged from 0.2383 (842–1061) to 1.2996 (525–1060), and the mean was 0.5684 ± 0.1575. A consensus tree of genetic relationships revealed two distinct groups (Figure 4), which coincided with those obtained from the STRUCTURE analysis (K = 2, 3; Figure 4). The populations in Group I (842, 843, 739, 754, 858, 1060, 1061, and 1565) were closely related but incompletely separated from the other populations; however, there was low support for this topology. The bootstrap probability values between the populations in Group I ranged from 0.23 (858–1060) to 1.00 (842–1061), and the mean value was 0.3838 ± 0.24. The bootstrap probability value between the populations in Group II ranged from 0.06 (307–102) to 0.78 (248–1869), and the mean value was 0.3483 ± 0.20. The approximately unbiased value was 0.45.

### 3.4. Population Differentiation in Phenotypic Traits and Q_ST_ Distance Matrices

Two-factor analyses of variance of the provenances and families within provenances were conducted for each trait in each year of the provenance trials (Appendix A). Highly significant differences were observed in 30 traits among all 22 provenances (*p* < 0.0001). Significant variation was observed in 29 traits among families within provenances (*p* < 0.0001); however, the *p*-value for CBH5 was only below 0.01. The significant variation observed in all 30 traits indicated that these populations were highly differentiated.

The *Q*_ST_ matrix was computed for the 30 traits across the five years of the provenance trials (Appendix A, [43]). The mean growth traits HEIT, CRB, and GBH/DBH increased from 1.544 to 10.697, from 0.919 to 2.820, and from 3.186 to 11.927, respectively. The mean SR, SMT, and NOB/CBH decreased from 0.803 to 0.546, from 1.730 to 1.416, and from 3.685 to 1.664, respectively. The annual mean of the *Q*_ST_ values for the HEIT increased from 0.321 to 0.499, while that for the NOB/CBH increased from 0.437 to 0.777. The annual mean of the *Q*_ST_ values for the CRB decreased from 0.488 to 0.461, while that of the GBH/DBH decreased from 0.556 to 0.497. However, the annual mean of the *Q*_ST_ values for the SMT ranged from 0.412 to 0.621, while that of the SR ranged from 0.508 to 0.743. Both of them sharply decreased in the second year (Figure 5).

### 3.5. Correlations between Q_ST_ Distance Matrices and The Geographic Distance Matrix

The *Q*_ST_ distance matrices for 15 of the 30 traits measured were significantly correlated with the geographic distance matrix (Appendix A; Table 4). The traits with significant correlations included the height, crown, DBH, and SMT traits. The correlation coefficients of the five height traits ranged from −0.338 to −0.237 (*p* < 0.001), the correlation coefficients of the four crown traits ranged from −0.373 to −0.157 (*p* < 0.0171), and the correlation coefficients of the three DBH traits and three SR traits ranged from −0.183 to 0.0210 (0.0272 < *p* < 0.001). These findings indicated that the *Q*_ST_ matrices of the HEIT, CRB, and SR were affected by geographic distance in five, four, and three of the years, respectively (Table 4). The *Q*_ST_ matrices of the other traits were less affected by geographic distance than the HEIT, CRB, and SR.

### 3.6. Analysis of Ecological Adaptation

#### 3.6.1. *Q*_ST_ Matrices of Traits Correlated with Climate Factors

The environmental factors were mostly negatively correlated with the *Q*_ST_ of the traits, and many of these correlations were highly significant (*p* < 0.001) (Appendix A). The HEIT was most strongly affected by the average temperature, average vapor pressure, and mean minimum temperature (*p* < 0.01 in more than four years). The GBH/DBH was most strongly affected by the wind direction with the maximum wind speed (*p* < 0.01 in all five years). The NOB/CBH was significantly affected by the extreme minimum temperature, average temperature, average vapor pressure, and mean maximum and minimum temperatures. The SMT was most strongly affected by the wind direction with the maximum wind speed (*p* < 0.05 in four years), average vapor pressure (*p* < 0.001 in three years), and average temperature (*p* < 0.01 in three years). The CRB was affected by the extreme maximum temperature and average station pressure in four years (*p* < 0.05). SR was most strongly affected by the minimum relative humidity in four years (*p* < 0.05); it was also affected by mean temperature anomalies in four years (*p* < 0.05). Temperature had a significant effect on all the traits and was thus the main cause of the differentiation in phenotypic traits.

#### 3.6.2. Seedling Survival

There was a significant decrease in seedling survivorship (54.6% survival) across the five years of the study. The mean *Q*_ST_ value in the second year was 0.508, which was significantly lower than that of the other years (0.732–0.743); this indicated that the pattern of local adaptation in the second year was different from that in the other years (Figure 6a). Three populations (739, 740, and 842) had substantial effects on the patterns of local adaptation, as their survival differed by as much as 52.3% compared with that of local populations (Figure 6b). None of the predictor variables explained the variance among population pairs in the patterns of local adaptation.

### 3.7. Q_ST_–F_ST_ Comparison

*Q*_ST_ and *F*_ST_ significantly differed for the maximum stem height, the straightness of the main trunk, and the survival rate (Table 5). The *Q*_ST_ values of HEIT 4, 5, and 6 were −0.0615, −0.0648, and 0.0711 lower than the *F*_ST_ values (*p* = 0.016, 0.0178, and 0.0064, respectively). The *Q*_ST_ value of SMT 6 was −0.0725 lower than the *F*_ST_ (*p* = 0.0072). The *Q*_ST_ values of SR 5, 6, 7, and 8 were 0.1047, 0.1016, 0.1254, and 0.1235 higher than the *F*_ST_ values (*p* = 0.0152, 0.0132, 0.0040, and 0.0042, respectively). The *Q*_ST_ values of growth traits were generally lower than the *F*_ST_ values, which indicated that the phenotypic differentiation was driven by uniform selection. The *Q*_ST_ value for the survival rate was higher than the *F*_ST_ value, which indicated divergent selection; this finding is consistent with the relationships among the populations inferred from the SSR data.

## 4. Discussion

We obtained strong evidence for the local adaptation *of M. azedarach* populations in Yunnan, Guizhou, Sichuan, and Gansu (Group I). The seedlings from southern Guizhou and northern Guangxi were the tallest and had the highest DBH and survival rates [56]. The families from Guangdong and Hainan increased rapidly in height, DBH, CBH, SMT, and CRB [57]. A sharp decrease in survival was observed for the Yunnan and Guizhou populations in the second year of the study (Figure 6). An analysis of 12 key traits of the fruit stones and seeds of 70 *M. azedarach* provenances collected from 17 provinces revealed that the geographic variation in the stones and seeds was affected by both longitude and latitude; however, the geographic variation was more pronounced with latitude than with longitude [58]. These findings suggest that climate is the most important factor shaping patterns of local adaptation in *M. azedarach*. Below, we provide a more detailed discussion of the main findings.

### 4.1. Detection of Local Adaptation

Significant correlations were observed between the *G*_ST_/(1 − *G*_ST_) for five loci and geographic distance. Patterns in the *G*_ST_ and *F*_ST_ revealed a significant but weak IBD effect (Table 3; Figure 2). An analysis of the genetic structure and genetic relationships using SSR loci revealed that the populations could be divided into two groups: Group I, comprising populations from the Yunnan and Guizhou provinces, and Group II, comprising two groups of populations separated by the Yangtze River. These findings are more informative than those derived from sequence-related amplified polymorphism (SRAP) analyses [57]. Consistent with the analysis of the geographic variation of the fruit stones and seeds, there was a southwest–northeast gradient in the width of the stones and seeds and a northwest–southeast gradient in the hundred-grain weight of the stones and seeds. Group I of the SSRs coincided with Groups III and IV of the stones and seeds [3,58].

Significant differences were detected in six growth traits in each of the five years according to an analysis of variance, which indicated substantial differentiation among populations. The *Q*_ST_ distance matrices of 15 growth traits were significantly correlated with the geographic distance matrix. The HEIT, GBH/DBH, NOB/CBH, SMT, CRB, and SR were affected by temperature in all five years, but the SR was most strongly affected by the minimum relative humidity in four of the years (*p* < 0.05). A sharp decrease in seedling survivorship to 54.6% was observed in the second year, and the mean *Q*_ST_ value was 0.508 in this year. This phenomenon has also been appeared in another trial [59]. Following a cold damage in Henan, the mortality rates of the southern seedlings from Lishui, Zhejiang province; Chengdu, Sichuan province; Liling, Hunan province; and Nanchang, Jiangxi province were 62.5%, 58.2%, 44.2%, and 41.7%, respectively. By contrast, the survival rate of the seedlings from northern China was greater than 95%.

According to twelve phenological indexes and three geographic factors, the distribution of *M. azedarach* in China has been divided into 11 phenological areas [60]. To facilitate provenance allocation, three areas of provenance transfer have been identified according to an analysis of the genetic structure using SSRs and SRAPs associated with adaptability at the seedling stage [58]. The flowering phenology of *M. azedarach*, such as the timing of various developmental stages, flowering, and fruiting, ranges from August in Bogor to September in Gambung, West Java [61]. All the phenological patterns of *M. azedarach* are indicative of local adaptation.

The patterns of genetic differentiation and growth trait differentiation, as well as variation in the phenology of *M. azedarach*, are consistent with local adaptation in this species. Climate is the most important factor shaping the patterns of population differentiation, and the temperature and minimum relative humidity play the most important roles in shaping patterns of local adaptation.

### 4.2. F_ST_–Q_ST_ Comparison in a Common Garden Experiment

We conducted a common garden experiment with a randomized block design, and a total of 11,055 individuals from all populations were used to estimate the *Q*_ST_. The *Q*_ST_ matrices of 30 traits significantly differed among the 22 provenances in a half-sib experiment (*p* < 0.0001). High sample sizes enhanced the accuracy of the *Q*_ST_ estimates [43]. The trend surface analysis of growth and seed morphological traits demonstrated that the phenotypes of chinaberry vary with latitude and longitude ([3,58]; Table 4). This variation in growth traits with latitude and longitude caused the *Q*_ST_ values to generally be significantly lower than the *F*_ST_ values, especially for the HEIT and SMT, which indicates uniform selection. This uniform selection on growth traits indicates the importance of rapid growth as a survival strategy in chinaberry, which is a pioneer tree species in forest ecosystems.

A population genetic differentiation analysis of *M. azedarach* revealed IBD and isolation by environment, and provided insights into the genetic relationships and genetic structure (Table 3; Figure 2 and Figure 4). Fossil studies suggest that these species might have originated during the Later Oligocene (30 million years ago (MYA)) in Nanning, China [62]. Other fruit fossils of *Melia* have been dated to the middle Miocene (20 MYA) of central Washington [63], the Middle Pleistocene (2 MYA) in northeastern Thailand [64], the Upper Miocene (24 MYA) in Poland [65], and the Holocene (1 MYA) in Jiangsu, China [66]. Geographic variation in the fruits and seeds of *M. azedarach* has been characterized in northern latitudes (18–38°) and eastern longitudes (100–122°) in China [58,67]. The size of seed fossils is similar to that of living species. Thus, the population with large seeds near the Sichuan Basin might have originated several MYA. The relationships among putative *M. toosedan* and putative *M. azedarach* were inferred by constructing a tree with 10,000 bootstrap replicates. The *Q*_ST_–*F*_ST_ distribution for each traittreatment combination was used to infer the empirical *p*-values associated with our point estimates of *Q*_ST_–*F*_ST_. The *Q*_ST_ of the SR was generally significantly higher than the *F*_ST_ at various latitudes, suggesting that natural selection has played a comparatively larger role in shaping the observed variation compared with genetic drift. The minimum relative humidity, average temperature, and geographic distance were the main environmental factors shaping local adaptation in *M. azedarach*.

## 5. Conclusions

Growth traits significantly varied with latitude and longitude in *M. azedarach*. The *Q*_ST_ of traits was calculated using a common garden experiment with a half-sib design. The *Q*_ST_ was significantly correlated with climate and geographic factors in all years. An analysis of the SSRs revealed significant population genetic differentiation. The survival rate was significantly affected by the population genetic differentiation, and patterns of local adaptation were affected by the minimum relative humidity, average temperature, and geographic distance. Generally, *Q*_ST_–*F*_ST_ comparisons are effective for clarifying patterns of local adaptation in *M. azedarach*.

## Figures and Tables

**Figure 1 genes-13-01924-f001:**
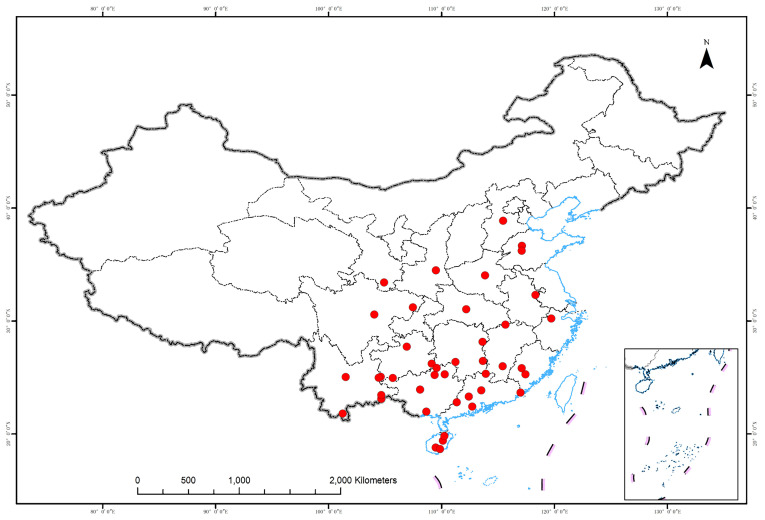
Geographic locations of the populations sampled in this study.

**Figure 2 genes-13-01924-f002:**
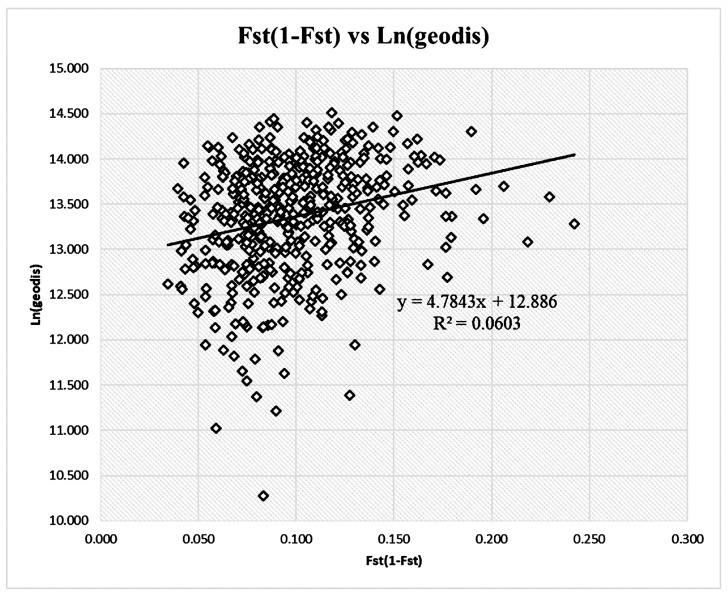
Mantel test of genetic differentiation according to population distances.

**Figure 3 genes-13-01924-f003:**
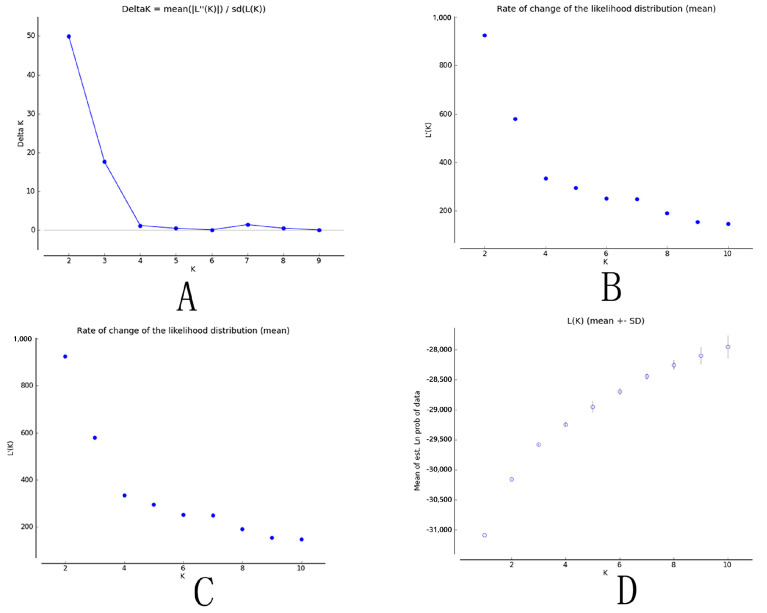
Scatter diagrams of the parameters from the STRUCTURE analysis. (**A**): ΔK; (**B**): L’(K); (**C**): L’’(K); and (**D**): Ln P(D).

**Figure 4 genes-13-01924-f004:**
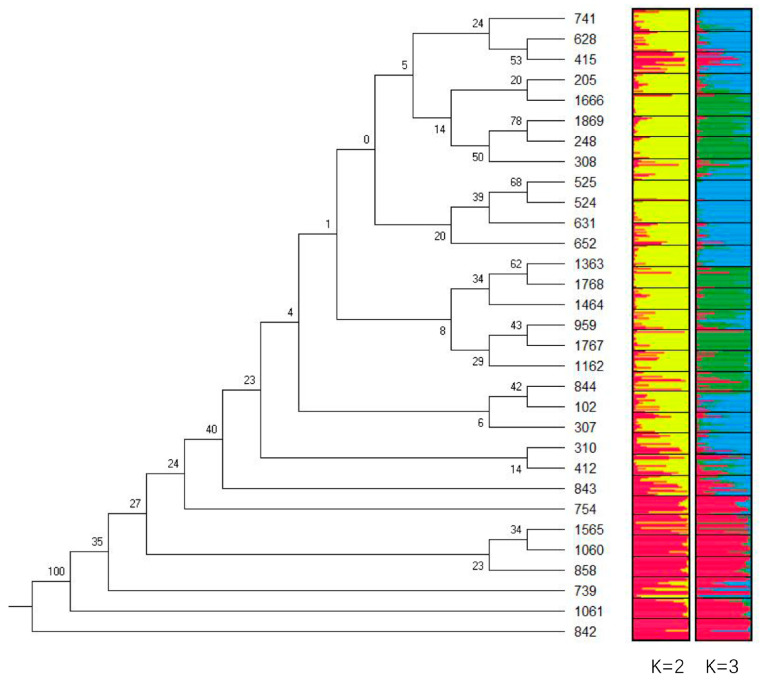
A consensus tree of the genetic relationships and genetic structure among 31 populations of *M. azedarach*. (Note: the numbers at the forks are the bootstrap probabilities estimated from 1000 bootstrap samples in the neighbor-joining method).

**Figure 5 genes-13-01924-f005:**
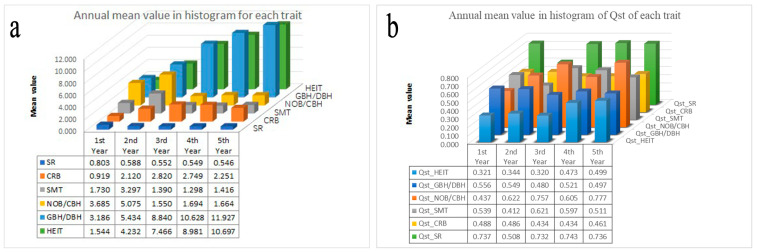
Annual mean value for each trait and the *Q*_ST_ of each trait during each year of the study.

**Figure 6 genes-13-01924-f006:**
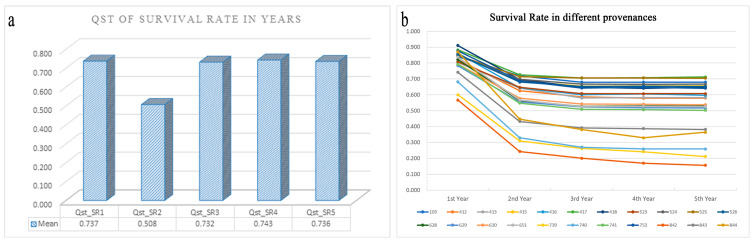
*Q*_ST_ mean value of the survival rate in each year (**a**); the survival rate of the different provenances in the five years (**b**).

**Table 1 genes-13-01924-t001:** Population information for this study.

Provenance Code ^a^	Experiments ^b^	Family Number ^c^	Sample Number	Province/Location	Latitude	Longitude	Altitude
412	I,II	16	15	Guangdong Renhua	25°19′	113°55′	99
415	I,II	20	15	Guangdong Kaiping	22°25′	112°43′	7
524	I,II	22	15	Hainan Tunchang	19°24′	110°07′	160
525	I,II	16	14	Hainan Wuzhishan	18°47′	109°29′	280
628	I,II	10	15	Guangxi Guilin	25°16′	110°17′	166
739 *	I,II	7	15	Yunnan Mengla	21°48′	101°15′	1010
741	I,II	11	15	Yunnan Malipo	23°06′	104°40′	1180
842 *	I,II	9	15	Guizhou Xingyi	25°03′	104°37′	1407
843 *	I,II	26	15	Guizhou Ceheng	24°57′	105°41′	1117
844	I,II	4	15	Guizhou Liping	26°13′	109°08′	618
102	I	-	15	Fujian Yong’an	25°49′	117°06′	255
205	I	-	15	Jiangxi Yudu	25°59′	115°25′	132
248	I	-	15	Jiangxi Ruichang	29°40′	115°40′	18
307	I	-	15	Hunan Dong’an	26°22′	111°14′	205
308	I	-	15	Hunan Liuyang	28°9′	113°38′	124
310	I	-	15	Hunan Yanling	26°27′	113°40′	200
631	I	-	15	Guangxi Qinzhou	21°58′	108°39′	17
652	I	-	15	Guangxi Du’an	23°55′	108° 6′	373
754 *	I	-	14	Yunnan Chuxiong	25°2′	101°31′	2173
858 *	I	-	15	Guizhou Zunyi	27°43′	106°55′	1168
959	I	-	15	Zhejiang Lin’an	30°13′	119°43′	47
1060 *	I	-	15	Sichuan Chengdu	30°34′	104° 3′	495
1061 *	I	-	15	Sichuan Dazhou	31°12′	107°28′	593
1162	I	-	15	Anhui Chuzhou	32°18′	118°19′	15
1363	I	-	15	Hubei Jingmen	31°2′	112°11′	98
1464	I	-	15	Shanxi Weinan	34°29′	109°30′	351
1565 *	I	-	14	Gansu Longnan	33°24′	104°55′	1106
1666	I	-	15	Hebei Baoding	38°52′	115°27′	22
1767	I	-	15	Shandong Jinan	36°39′	117°7′	122
1768	I	-	15	Shandong Tai’an	36°13′	117°6′	641
1869	I	-	14	Henan Xuchang	34°2′	113°51′	71
103	II	1	-	Fujian Zhangping	25°16′	117°26′	219
413	II	9	-	Guangdong Yunan	22°48′	111°21′	22
416	II	7	-	Guangdong Qingyuan	23°51′	113°31′	73
417	II	4	-	Guangdong En’ping	23°18′	112°25′	17
418	II	3	-	Guangdong Raoping	23°39′	117°00′	20
523	II	8	-	Hainan Haikou	19°49′	110°15′	129
526	II	9	-	Hainan Lingshui	18°39′	109°52′	79
629	II	18	-	Guangxi Rong’an	25°13′	109°23′	226
630	II	19	-	Guangxi Sanjiang	25°50′	109°34′	240
651	II	10	-	Guangxi Qinzhou	21°58′	108°39′	250
740	II	3	-	Yunnan Luoping	24°58′	104°26′	1415
753	II	4	-	Yunnan Xichou	23°26′	104°40′	1217

Note: ^a^, * putative *Melia toosendan*, others are *M azedarach*; ^b^, I, population used in SSR analysis; II, population used in common garden; ^c^, ‘-’, the families were not used in common garden, or the samples were not used in the SSR analysis.

**Table 2 genes-13-01924-t002:** Number of alleles, polymorphic information content (PIC), expected heterozygosity (He), gene diversity (h), and Shannon’s information index (*I*) for 31 populations of *M. azedarach* L.

Population	Na (±Sd)	Range	Ne (±Sd)	Range	He (±Sd)	H (±Sd)	I_Shannon’s	PIC
412	8.07 ± 2.71	(4, 12)	5.24 ± 1.64	(2.99, 8.03)	0.82 ± 0.07	0.79 ± 0.07	1.78 ± 0.34	0.761
741	7.53 ± 2.56	(4, 12)	5.33 ± 1.84	(2.96, 8.33)	0.82 ± 0.07	0.79 ± 0.07	1.75 ± 0.36	0.759
631	6.40 ± 2.41	(4, 10)	4.61 ± 1.90	(2.26, 7.04)	0.77 ± 0.12	0.74 ± 0.11	1.57 ± 0.44	0.698
739	8.00 ± 2.62	(4, 11)	5.70 ± 2.29	(2.47, 9.78)	0.82 ± 0.09	0.79 ± 0.09	1.80 ± 0.41	0.763
205	7.20 ± 2.51	(4, 13)	5.06 ± 1.77	(3.02, 8.65)	0.81 ± 0.07	0.78 ± 0.07	1.71 ± 0.33	0.749
524	6.60 ± 2.56	(3, 11)	4.56 ± 1.73	(2.49, 7.75)	0.78 ± 0.10	0.75 ± 0.09	1.58 ± 0.40	0.709
525	6.00 ± 2.93	(3, 11)	4.09 ± 1.86	(2.27, 8.45)	0.74 ± 0.11	0.72 ± 0.10	1.46 ± 0.43	0.666
628	7.60 ± 2.72	(4, 12)	5.29 ± 1.99	(2.79, 8.82)	0.81 ± 0.08	0.78 ± 0.08	1.74 ± 0.38	0.752
415	6.87 ± 2.36	(4, 11)	4.58 ± 1.56	(2.43, 7.89)	0.78 ± 0.09	0.76 ± 0.09	1.62 ± 0.37	0.718
307	7.20 ± 2.39	(4, 12)	5.30 ± 2.05	(2.28, 9.18)	0.81 ± 0.10	0.78 ± 0.10	1.72 ± 0.41	0.746
842	6.53 ± 2.83	(2, 12)	4.56 ± 2.09	(2.00, 8.82)	0.77 ± 0.11	0.74 ± 0.10	1.56 ± 0.45	0.696
308	7.87 ± 2.47	(5, 14)	5.36 ± 1.33	(3.24, 7.26)	0.83 ± 0.06	0.80 ± 0.06	1.80 ± 0.29	0.773
310	7.80 ± 2.14	(5, 12)	5.33 ± 1.81	(2.92, 9.00)	0.82 ± 0.07	0.79 ± 0.06	1.78 ± 0.31	0.764
843	8.60 ± 3.44	(4, 14)	5.78 ± 2.62	(2.84, 10.22)	0.82 ± 0.09	0.79 ± 0.09	1.81 ± 0.46	0.760
102	6.93 ± 2.21	(3, 11)	4.83 ± 1.42	(2.38, 6.61)	0.80 ± 0.08	0.77 ± 0.08	1.67 ± 0.33	0.738
844	7.87 ± 2.64	(5, 12)	5.55 ± 2.18	(2.82, 9.18)	0.82 ± 0.08	0.79 ± 0.08	1.79 ± 0.38	0.764
1061	8.07 ± 2.79	(3, 14)	5.51 ± 2.28	(2.26, 10.47)	0.81 ± 0.10	0.78 ± 0.10	1.78 ± 0.44	0.751
1767	6.87 ± 2.17	(4, 10)	5.05 ± 1.71	(2.74, 8.18)	0.81 ± 0.08	0.78 ± 0.08	1.69 ± 0.35	0.746
754	6.73 ± 2.99	(3, 13)	4.76 ± 2.38	(2.11, 10.56)	0.77 ± 0.11	0.75 ± 0.10	1.59 ± 0.46	0.704
959	6.60 ± 2.29	(4, 12)	4.78 ± 1.95	(2.59, 9.78)	0.79 ± 0.09	0.76 ± 0.09	1.62 ± 0.36	0.723
1565	7.20 ± 1.70	(4, 10)	4.94 ± 1.40	(2.99, 7.68)	0.81 ± 0.06	0.78 ± 0.06	1.71 ± 0.27	0.750
1666	7.60 ± 2.41	(3, 13)	5.18 ± 1.61	(2.38, 7.76)	0.81 ± 0.08	0.79 ± 0.08	1.75 ± 0.34	0.754
1768	7.60 ± 2.10	(4, 11)	5.29 ± 1.76	(2.60, 9.33)	0.82 ± 0.08	0.79 ± 0.07	1.76 ± 0.33	0.759
1464	7.40 ± 2.61	(4, 13)	5.10 ± 1.83	(3.49, 8.49)	0.81 ± 0.06	0.78 ± 0.06	1.72 ± 0.33	0.752
1363	7.33 ± 2.49	(3, 11)	5.14 ± 1.86	(2.51, 8.05)	0.81 ± 0.10	0.78 ± 0.09	1.71 ± 0.40	0.742
652	7.60 ± 1.76	(5, 11)	5.32 ± 1.87	(3.23, 9.56)	0.82 ± 0.06	0.79 ± 0.06	1.76 ± 0.29	0.763
1869	7.13 ± 2.50	(4, 12)	4.71 ± 1.70	(2.60, 8.34)	0.79 ± 0.08	0.76 ± 0.08	1.64 ± 0.36	0.725
1060	7.27 ± 2.66	(4, 12)	4.99 ± 2.25	(2.67, 10.00)	0.79 ± 0.10	0.76 ± 0.10	1.67 ± 0.43	0.724
1162	7.60 ± 2.06	(5, 11)	5.19 ± 1.58	(3.21, 8.90)	0.82 ± 0.05	0.79 ± 0.05	1.76 ± 0.27	0.764
858	6.73 ± 2.87	(3, 14)	4.71 ± 2.16	(2.13, 10.00)	0.77 ± 0.11	0.74 ± 0.11	1.59 ± 0.45	0.703
248	6.67 ± 1.50	(4, 9)	4.58 ± 1.07	(2.76, 6.42)	0.80 ± 0.06	0.77 ± 0.06	1.63 ± 0.24	0.734
Mean	7.27 ± 2.46	□	5.05 ± 1.85	□	0.80 ± 0.08	0.77 ± 0.08	1.69 ± 0.39	0.739

**Table 3 genes-13-01924-t003:** Population genetic differentiation and tests of IBD at individual SSR loci in *M. azedarach*.

Locus	*G*st	a	b	*p*-Value	r
SSR02	0.0731	0.0721	−0.0023	0.300	0.0400
SSR29	0.0915	0.0580	−0.0003	0.520	0.0032
SSR54	0.1010	−0.2353	0.0221	0.010	0.2625
SSR59	0.0687	0.0856	−0.0035	0.240	0.0640
SSR74	0.1230	−0.1200	0.0145	0.030	0.1552
SSR111	0.0623	−0.1486	0.0138	0.010	0.2311
SSR113	0.1017	0.0060	0.0042	0.260	0.0374
SSR114	0.1064	0.0920	−0.0021	0.330	0.0224
SSR116	0.1097	−0.0553	0.0091	0.060	0.1100
SSR117	0.1281	−0.1173	0.0147	0.050	0.1265
SSR118	0.0762	0.0537	−0.0008	0.460	0.0141
SSR119	0.0448	0.0125	0.0009	0.350	0.0283
SSR120	0.0327	0.0065	0.0008	0.280	0.0332
SSR122	0.0866	−0.0019	0.0039	0.190	0.0608
SSR123	0.0984	−0.2884	0.0260	0.010	0.3056
Multilocus	0.0860	−0.0353	0.0063	0.010	0.2449

Note: all multilocus *G*_ST_ values significantly differed from zero (*p* < 0.001). In *G*_ST_/(1 − *G*_ST_) = a + b ln(geographic distance), a and b refer to the intercept and regression coefficient, respectively; r refers to the Pearson’s correlation coefficient between *G*_ST_/(1 − *G*st) and geographic distance.

**Table 4 genes-13-01924-t004:** Correlation between the *Q*_ST_ matrices of the 30 measured traits and the geographic distance matrix by year.

	*Q*st_HEIT	*Q*st_GBH/DBH	*Q*st_NOB/CBH	*Q*st_SMT	*Q*st_CRB	*Q*st_SR
1st Year	−0.3219 ***	−0.0190 ^ns^	−0.1024 ^ns^	−0.2176 ***	−0.1937 ***	−0.1832 **
2nd Year	−0.2675 ***	−0.1524 *	0.0105 ^ns^	−0.0493 ^ns^	−0.0896 ^ns^	0.0746 ^ns^
3rd Year	−0.3327 ***	−0.1454 *	−0.0447 ^ns^	−0.1558 *	−0.1567 *	0.0210 **
4th Year	−0.2367 ***	−0.1111 ^ns^	−0.0760 ^ns^	−0.1161 ^ns^	−0.3726 ***	−0.1704 **
5th Year	−0.3384 ***	−0.1736 **	−0.1795 **	−0.0952 ^ns^	−0.1793 **	−0.1193 ^ns^

Note: ‘^ns^’, ‘*’, ‘**’, and ‘***’ mean that the *p*-value of the *Q*st is greater than 0.05; greater than 0.01 and less than 0.05; greater than 0.001 and less than 0.01; and less than 0.001, respectively.

**Table 5 genes-13-01924-t005:** Comparison of the *Q*_ST_ values of traits with the *F*_ST_ values of microsatellite alleles in different populations.

Trait Code	Qst–Fst	Lower Bound Crit. Value. 2.5%	Upper Bound Crit. Value. 97.5%
HEIT1	−0.0615 *	−0.0529	0.0720
HEIT2	−0.0648 *	−0.0565	0.0833
HEIT3	−0.0711 **	−0.0565	0.0815
HEIT4	−0.0484 ^ns^	−0.0578	0.0885
HEIT5	−0.0330 ^ns^	−0.0597	0.0917
GBH1	0.0306 ^ns^	−0.0540	0.0713
DBH2	−0.0127 ^ns^	−0.0567	0.0813
DBH3	−0.0120 ^ns^	−0.0558	0.0773
DBH4	−0.0072 ^ns^	−0.0566	0.0803
DBH5	−0.0285 ^ns^	−0.0557	0.0792
NOB1	−0.0112 ^ns^	−0.0533	0.0699
NOB2	−0.0112 ^ns^	−0.0533	0.0699
CBH3	−0.0120 ^ns^	−0.0558	0.0773
CBH4	0.0329 ^ns^	−0.0817	0.3663
CBH5	0.0789 ^ns^	−0.0859	0.4238
SMT1	−0.0156 ^ns^	−0.0593	0.0984
SMT2	−0.0591 ^ns^	−0.0609	0.0999
SMT3	−0.0725 **	−0.0584	0.0937
SMT4	−0.0639 ^ns^	−0.0647	0.1297
SMT5	−0.0477 ^ns^	−0.0613	0.1001
CRB1	−0.0179 ^ns^	−0.0540	0.0753
CRB2	0.0041 ^ns^	−0.0586	0.0929
CRB3	−0.0081 ^ns^	−0.0574	0.0925
CRB4	−0.0099 ^ns^	−0.0571	0.0873
CRB5	−0.0245 ^ns^	−0.0573	0.0866
SR1	0.0350 ^ns^	−0.0569	0.0825
SR2	0.1047 *	−0.0557	0.0790
SR3	0.1016 *	−0.0551	0.0763
SR4	0.1254 **	−0.0558	0.0756
SR5	0.1235 **	−0.0536	0.0746

Note: ‘^ns^’, ‘*’, and ‘**’ mean the two-tailed *p* value of ‘Qst–Fst’ is greater than 0.05; greater than 0.01 and less than 0.05; greater than 0.001 and less than 0.01, respectively.

## Data Availability

Not applicable.

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
