# Peer review of "Climate-Driven Adaptive Differentiation in Melia azedarach: Evidence from a Common Garden Experiment"

_genes, 2022, doi:10.3390/genes13111924_

Round 1
Reviewer 1 Report
Comments and Suggestions for Authors
The study carried out is aimed to determine whether there is local adaptation among Melia azedarach populations according to analyses of phylogenetic relationships and population structure derived from molecular markers and whether ecological adaptation plays a role in species differentiation through provenance trials. M. azedarach is an economically important timber species and highly variable in phenotype across its range in China. Study of local adaptation in its populations is important for clarifying patterns of population differentiation of this species across its natural range. The features often used to distinguish Melia species, such as the genes underlying differences in phytochemicals and phylogenetic relationships among species, have not been thoroughly explored. Patterns of genetic differentiation among populations were studied using molecular markers (simple sequence repeats) and data on phenotypic variation in six traits. The results of these analyses provide new insights into the role of natural selection in shaping patterns of differentiation among M. azedarach populations.
The manuscript is composed according to the requirements of “Genes”. The research methods applied are appropriate and sufficient to achieve the objectives of the study. The results are well presented and supported by tables and figures that are of good quality.
The following recommendations can be made:
Abstract:
Similar to the entered meaning of the abbreviation QST, to give the meaning of and of FST (Line 29).
Introduction:
The sentences: Lines 90 - 91“We conducted a common garden experiment and used molecular markers to clarify patterns of differentiation”, Lines 108 – 110: We used highly polymorphic nuclear DNA satellite markers to evaluate the role of other evolutionary mechanisms (genetic drift, gene flow, and mutation) in lineage sorting and patterns of differentiation in the M. azedarach complex.among members of the M. azedarach complex.” and Lines 125 - 127“We obtained samples of M. azedarach from 43 populations; samples from 22 of these populations were used in a common garden experiment, and samples from 31 populations were used in analyses of SSR markers.” are more appropriate for “Material and methods”
Results
In the Table 2 it is not necessary to mark the corresponding values described in the text, it is sufficient to cite the table.
The note below Table 3 must be separated from the rest of the text by a blank line
A note to be given for Table 4 to explain what the numbers in blue show. I think it's better tomark them with an asterisk or a letter. Also, since obviously in every second row after the main designation in the first column (1st year, 2nd year, …etc.) are represented the P-values, this second row can be denoted as “P-value” Like in Table 2, it is not necessary to mark the values, it's enough to comment them in the text.
Table 5: The values marked in yellow and red respectively to be signed in another way (with asterisk or letter) and in a note below the table give their meaning
Author Response
10th Oct. 2022
Ms. Catherine WangSection Managing Editor
genes
Dear Editor and reviewers,
Thanks for your comments and suggestions for the manuscript entitled “Climate-driven adaptive differentiation in Melia azedarach: evidence from a common garden experiment,” which we submitted to genes on the 18th of September.
We revised this manuscript according to these comments. Point-by-point responses are given below.
Responses to the Reviewer 1:
Comments and Suggestions for Authors
The study carried out is aimed to determine whether there is local adaptation among Melia azedarach populations according to analyses of phylogenetic relationships and population structure derived from molecular markers and whether ecological adaptation plays a role in species differentiation through provenance trials. M. azedarach is an economically important timber species and highly variable in phenotype across its range in China. Study of local adaptation in its populations is important for clarifying patterns of population differentiation of this species across its natural range. The features often used to distinguish Melia species, such as the genes underlying differences in phytochemicals and phylogenetic relationships among species, have not been thoroughly explored. Patterns of genetic differentiation among populations were studied using molecular markers (simple sequence repeats) and data on phenotypic variation in six traits. The results of these analyses provide new insights into the role of natural selection in shaping patterns of differentiation among M. azedarach populations.
The manuscript is composed according to the requirements of “Genes”. The research methods applied are appropriate and sufficient to achieve the objectives of the study. The results are well presented and supported by tables and figures that are of good quality.
Answer: Thanks for your positive comments on this manuscript.
The following recommendations can be made:
Abstract:
Similar to the entered meaning of the abbreviation QST, to give the meaning of and of FST (Line 29).
Answer: Thanks for your suggestion, the meaning of the abbreviation of Fst was added in line 29.
Introduction:
The sentences: Lines 90 - 91“We conducted a common garden experiment and used molecular markers to clarify patterns of differentiation”, Lines 108 – 110: We used highly polymorphic nuclear DNA satellite markers to evaluate the role of other evolutionary mechanisms (genetic drift, gene flow, and mutation) in lineage sorting and patterns of differentiation in the M. azedarach complex.among members of the M. azedarach complex.” and Lines 125 - 127“We obtained samples of M. azedarach from 43 populations; samples from 22 of these populations were used in a common garden experiment, and samples from 31 populations were used in analyses of SSR markers.” are more appropriate for “Material and methods”
Answer: Thanks for your suggestion, these sentences in lines 90-91, 108-110, and 125-127 were revised. We checked the content in the Material and Methods.
Results
In the Table 2 it is not necessary to mark the corresponding values described in the text, it is sufficient to cite the table.
Answer: Thanks for your suggestion. We removed the marks.
The note below Table 3 must be separated from the rest of the text by a blank line
Answer: Thanks for your suggestion. Table 3 was kept a blank line from the rest of the text.
A note to be given for Table 4 to explain what the numbers in blue show. I think it's better tomark them with an asterisk or a letter. Also, since obviously in every second row after the main designation in the first column (1st year, 2nd year, …etc.) are represented the P-values, this second row can be denoted as “P-value” Like in Table 2, it is not necessary to mark the values, it's enough to comment them in the text.
Answer: Thanks for your comments. The second row was removed. And the p-value was denoted by ‘ns’, ‘*’, ‘**’, and ‘***’ at the top right corner of the Qst values. The revised Table 4 and notice were added to the text in lines 359-361.
Table 5: The values marked in yellow and red respectively to be signed in another way (with asterisk or letter) and in a note below the table give their meaning
Answer: Thanks for your suggestion, the color marked in the tables was removed. And the p-value was indicated under table 5. The original data of the two-tailed p value were removed. Responses to the editors:
In line 147-148: For all the figures, please use high resolution picture (at least 300 dpi). We recommend that it is best to provide sharper tif/jpg files in order to produce a clear pdf file.
Answer: Thanks for your suggestion. All images are individually modified to the resolution of more than 300 dpi in a single file.
In line 274-275: It is recommended to use p to unify the full text. The same goes for others.
Answer: Thanks for your suggestion. All the p values were revised to a consistent type.
In line 285-286: Are these bold necessary? If so, please explain in the table footer; if not, please remove.
Answer: Thanks for your suggestion. These bolds were removed from the table and others.
In line 346-347: Please combine all subgraphs into one picture. If there is text, embed it in the picture instead of inserting a text box in word.
Answer: Thanks for your suggestion. All subgraphs were combined into one picture and the text was embed into it.
In line 359-360: Are the background color necessary in the table? If so, please explain in the table footer; if not, please remove.
Answer: Thanks for your suggestion. The background color in Table 4 was removed.
In line 388-389: Please combine all subgraphs into one picture. If there is text, embed it in a picture instead of inserting a text box in word.
Answer: Thanks for your suggestion. All subgraphs were combined into one picture and the text was embed into it.
In line 507: Please check if this is correct.
Answer: Thanks for your reminds. The grant numbers for this funding were added for this study.
We checked all the data from the original six traits, and added table S8 and table S9. All the Tables were checked again.
Thanks for your suggestion.
Sincerely,
Boyong Liao
Zhongkai University of Agriculture and Engineering
Phone No: +86 15920337317
E-mail Address: liaoby05@126.com

Reviewer 2 Report
This manuscript leaves a very favorable impression. The presented results are relevant and interesting to the scientific community. However, there are a number of remarks:
1. In general, a lot of work has been done on molecular markering, but the results have not been properly analyzed. SSR-markers can show linkage with particular traits. This aspect has not been analyzed. And it even suggests itself from the goals and provided that it is said about 6 traits.
2. According to the results obtained, the most variable loci can also be identified. It is necessary to insert information on the number of alleles for each of the loci. And for further marking, not the entire set of markers is recommended, but only the most variable ones.
3. Maybe I'm not careful, but I didn't see the original data on 6 phenotypic traits that were analyzed.
4. Figures 3 and 6 are too small and unreadable. Figure 5 is also hard to read.
Author Response
10th Oct. 2022
Ms. Catherine WangSection Managing Editor
genes
Dear Editor and reviewers,
Thanks for your comments and suggestions for the manuscript entitled “Climate-driven adaptive differentiation in Melia azedarach: evidence from a common garden experiment,” which we submitted to genes on the 18th of September.
We revised this manuscript according to these comments. Point-by-point responses are given below.
Responses to the Reviewer 2:
Comments and Suggestions for Authors
This manuscript leaves a very favorable impression. The presented results are relevant and interesting to the scientific community. However, there are a number of remarks:
- In general, a lot of work has been done on molecular markering, but the results have not been properly analyzed. SSR-markers can show linkage with particular traits. This aspect has not been analyzed. And it even suggests itself from the goals and provided that it is said about 6 traits.
Answer: Thanks for your suggestions. For these 15 SSRs, it is weak in numbers to analyze the linkage with the six traits observed. SNP from RAD-seq, resequencing and transcriptome data will be comprehensively used in association analysis of Chinaberry tree in the next manuscript. Thanks for your suggestion again.
- According to the results obtained, the most variable loci can also be identified. It is necessary to insert information on the number of alleles for each of the loci. And for further marking, not the entire set of markers is recommended, but only the most variable ones.
Answer: Thanks for those suggestions. In Table S1, the number of alleles was listed for each locus. These limited SSRs were screened from the related species of chinaberry in published papers. For screening for the most variable SSR loci, we would carry it out in the data of genome-wide analysis in next year.
- Maybe I'm not careful, but I didn't see the original data on 6 phenotypic traits that were analyzed.
Answer: Thanks for your suggestion. The original data on 6 phenotypic traits were added in Table S8.
- Figures 3 and 6 are too small and unreadable. Figure 5 is also hard to read.
Answer: Thanks for your suggestion. The quality of Figure 3, 5, and 6 was revised.
Responses to the editors:
In line 147-148: For all the figures, please use high resolution picture (at least 300 dpi). We recommend that it is best to provide sharper tif/jpg files in order to produce a clear pdf file.
Answer: Thanks for your suggestion. All images are individually modified to the resolution of more than 300 dpi in a single file.
In line 274-275: It is recommended to use p to unify the full text. The same goes for others.
Answer: Thanks for your suggestion. All the p values were revised to a consistent type.
In line 285-286: Are these bold necessary? If so, please explain in the table footer; if not, please remove.
Answer: Thanks for your suggestion. These bolds were removed from the table and others.
In line 346-347: Please combine all subgraphs into one picture. If there is text, embed it in the picture instead of inserting a text box in word.
Answer: Thanks for your suggestion. All subgraphs were combined into one picture and the text was embed into it.
In line 359-360: Are the background color necessary in the table? If so, please explain in the table footer; if not, please remove.
Answer: Thanks for your suggestion. The background color in Table 4 was removed.
In line 388-389: Please combine all subgraphs into one picture. If there is text, embed it in a picture instead of inserting a text box in word.
Answer: Thanks for your suggestion. All subgraphs were combined into one picture and the text was embed into it.
In line 507: Please check if this is correct.
Answer: Thanks for your reminds. The grant numbers for this funding were added for this study.
We checked all the data from the original six traits, and added table S8 and table S9. All the Tables were checked again.
Thanks for your suggestion.
Sincerely,
Boyong Liao
Zhongkai University of Agriculture and Engineering
Phone No: +86 15920337317
E-mail Address: liaoby05@126.com

Round 2
Reviewer 2 Report
The submitted manuscript has improved significantly. However, the results of the analysis still raise questions. Since you write that many of the analyzes are not fully completed, and the results are expected to be received only by next year, the question arises whether this study should be published now. The results of genetic analysis, it seems, have not yet been fully processed. However, this does not diminish the significance of what has already been done and this manuscript, at the discretion of the editor, can be published in the absence of other claims.